# The Evaluation of Drugs as Potential Modulators of the Trafficking and Maturation of ACE2, the SARS-CoV-2 Receptor

**DOI:** 10.3390/biom14070764

**Published:** 2024-06-27

**Authors:** Nesreen F. Alkhofash, Bassam R. Ali

**Affiliations:** 1Department of Genetics and Genomics, College of Medicine and Health Sciences, United Arab Emirates University, Al-Ain P.O. Box 15551, United Arab Emirates; nesreenfrk@hotmail.com; 2ASPIRE Precision Medicine Research Institute Abu Dhabi, United Arab Emirates University, Al-Ain P.O. Box 15551, United Arab Emirates

**Keywords:** ACE2, COVID-19, protein trafficking, drugs screening, cellular localization, molecular modulators of transportation

## Abstract

ACE2, part of the angiotensin-converting enzyme family and the renin–angiotensin–aldosterone system (RAAS), plays vital roles in cardiovascular and renal functions. It is also the primary receptor for SARS-CoV-2, enabling its entry into cells. This project aimed to study ACE2’s cellular trafficking and maturation to the cell surface and assess the impact of various drugs and compounds on these processes. We used cellular and biochemical analyses to evaluate these compounds as potential leads for COVID-19 therapeutics. Our screening assay focused on ACE2 maturation levels and subcellular localization with and without drug treatments. Results showed that ACE2 maturation is generally fast and robust, with certain drugs having a mild impact. Out of twenty-three tested compounds, eight significantly reduced ACE2 maturation levels, and three caused approximately 20% decreases. Screening trafficking inhibitors revealed significant effects from most molecular modulators of protein trafficking, mild effects from most proposed COVID-19 drugs, and no effects from statins. This study noted that manipulating ACE2 levels could be beneficial or harmful, depending on the context. Thus, using this approach to uncover leads for COVID-19 therapeutics requires a thorough understanding ACE2’s biogenesis and biology.

## 1. Introduction

The global pandemic of COVID-19 caused by the novel coronavirus, known as SARS-CoV-2, inflicted unprecedented economic, societal, and health burdens worldwide. Crucially, similar to SARS-CoV and unlike MERS-CoV, the spike S protein of the SARS-CoV-2 virus mediates the viral attachment and entry into host cells by binding to its target receptor, the angiotensin-converting enzyme 2, ACE2 [1]. The spike S protein of SARS-CoV-2 binds with high affinity and specificity to ACE2, and this binding affinity is about 20 times higher than that reported for SARS-CoV [2]. Such high specificity and affinity for ACE2 may explain the apparent ease with which the virus spreads and transmits between humans, explaining some of its high contagiousness. Additionally, ACE2’s wide tissue distribution in the human body may explain the multi-systemic nature of the clinical symptoms of the SARS-CoV-2 infection. Although most people are susceptible to this viral infection, the nature and severity of the disease varies significantly from one person to another and across populations [3,4]. That was evident from the high variability in disease burden and fatality rates between countries and populations [3]. It was suggested that one possible contributing factor might be the variable expression levels of ACE2 among individuals [5]. Therefore, we wondered if the cellular trafficking maturation of ACE2 could be manipulated by altering its cell surface expression levels as a potential lead target for treating the COVID-19 infection or reducing its severity [6].

Aside from its role as a SARS-CoV-1 and SARS-CoV-2 receptor, ACE2 is a member of the renin–angiotensin system (RAS) and plays a role in regulating blood pressure and fluid and electrolyte homeostasis [7]. The ACE2 gene is located on the X chromosome, escapes X-inactivation, and encodes an 805 amino acid membrane-bound glycoprotein with a carboxypeptidase activity [8]. The receptor consists of a carboxy-terminal domain, an N-terminal peptide region, and a catalytic domain with zinc-binding metalloprotease motif [9]. Its C-terminal region is homologous with collectrin, a developmentally regulated renal protein with no catalytic domain and no similarity with ACE [8].

As already indicated, ACE2’s binding affinity with the spike S protein has been shown to be one of the most important factors determining the infectivity of SARS-CoV viruses. In fact, the S protein of SARS-CoV-2 binds specifically to ACE2 and not to any previously reported coronavirus receptor, such as the aminopeptidase N or the dipeptidyl peptidase 4 (DPP4) that are used by the MERS-CoV virus [10]. ACE2 knock-out studies revealed that this protein is crucial for the SARS-CoV viral infection, which consequently downregulates its expression, leading to severe acute respiratory failure in a wild-type animal model [11]. Due to its crucial biological functions and interactions with viruses, ACE2 is considered a double-edged sword in relation to the treatment of COVID-19 infection, although the exact pathways and roles are not very well defined. Humanized transgenic animal modeling further demonstrated that human ACE2 specifically mediates host cell invasion by SARS-CoV2 [12]. Due to the recent COVID-19 pandemic, numerous research activities have been conducted and reported on various therapeutic approaches targeting different stages of the virus’ life cycle and critical components required for the infection, including the various aspects of ACE2 homeostasis including its interactions, availability, and trafficking [13,14,15,16,17,18]

One mechanism to downregulate ACE2 could be through the shedding of ACE2 ectodomain into circulation, producing an ACE2 soluble form with preserved catalytic activity [19], which has been shown to decrease the viral infection efficiency with inhibitory effects in the airways [20]. Another mechanism to downregulate ACE2 could be by directing ACE2 to degradation, with one way being by increasing the levels of Ang II, causing ACE2 internalization and lysosomal degradation [21]. Furthermore, co-immunoprecipitation experiments demonstrated that AT1R and ACE2 form complexes and these interactions are decreased by Ang II treatment, enhancing ACE2 ubiquitination and marking it for proteasomal degradation [21]. Overall, ACE2 turnover follows both the proteasomal and lysosomal degradation routes.

Yet another potential target in ACE2 trafficking is to manipulate its trafficking and exit from the ER. Similar to ACE, ACE2 is an N-glycosylated protein that is transported from the ER to the Golgi apparatus and then to the cell surface by vesicular transport. N-glycosylation of ACE2 occurs at different amino acid residues initially within the endoplasmic reticulum (ER), which are further processed in the Golgi apparatus to produce the fully mature glycoprotein at the cell surface. As mentioned, characterizing the interactions between ACE2 and other proteins and understanding their role during intracellular trafficking might represent a potential therapeutic target for SARS-CoV-2 as well as other coronavirus infections by decreasing ACE2 availability at the cell surface. However, we should take into consideration the protective effect of ACE2 against Ang II-induced deleterious effects because the overactivation of the Angiotensin (Ang) II/AT1R axis may explain the multiorgan dysfunction seen in COVID-19 patients with acute and chronic inflammation [22].

In this article, we report the development of a cellular assay to evaluate the trafficking and maturation levels of ACE2 as indicators for its cell surface expression. In addition, we employed this assay to evaluate the potential impact of 23 drugs and compounds on its maturation. The tested compounds were selected from the literature, which suggested that the compounds were relevant to COVID-19 infection, protein trafficking, or viral biology, etc. Overall, our data re-emphasize the robustness of ACE2 trafficking and maturation competency and its resilience to disruption [23]. ACE2 was trafficked to the plasma membrane fairly quickly and semi-quantitively. The majority of the tested drugs showed no or little impact on ACE2 trafficking out of the ER to the plasma membrane, as evidenced by the high maturation of its N-glycans. However, out of the 23 tested compounds, eight showed a low but significant decrease in ACE2 maturation with a *p* value less than 0.05, three showed a ~20% decrease, while Brefeldin A (BFA), as expected, showed a complete block of ACE2 maturation. Statins, on the other hand, showed no effects.

## 2. Materials and Methods

### 2.1. Cell Lines and Cell Culture

The HeLa and HEK293T cells used in this study were obtained from ATCC, and both were cultured in Dulbecco’s Modified Eagle Medium (Gibco, Grand Island, NY, USA) as previously described [23]. Culture media were supplemented with 10% fetal bovine serum (Gibco) and antibiotic–antimycotic (Gibco), and cells were incubated at 37 °C with 5% CO_2_.

### 2.2. Compounds and Drugs Screened for Inhibition of ACE2 Maturation

Table 1 lists the drugs and compounds that were selected and screened for their potential impact on ACE2 cellular trafficking and maturation. The volume of DMSO used as a control for each experiment was always equivalent to the volume used for the drug-treated samples. In all cases, the percentage of DMSO never exceeded 0.4%. The starting optimization concentrations for these compounds were derived from the literature (Appendix A).

### 2.3. Immunofluorescence and Confocal Microscopy

HeLa cells obtained from ATCC were seeded on sterilized coverslips in 24-well plates. At 60% confluency, cells were co-transfected with WT ACE2 and GFP-tagged H-Ras plasmid; the GFP-HRas is a well-established plasma membrane marker [23]. The expressed ACE2 protein was derived from the cDNA of isoform 2 (Uniprot ID Q9BYF1) cloned into the used mammalian expression vector (NM_021804; OriGene RC208442). Therefore, we did not expect to see the expression of the other ACE2 isoforms. After the specified expression time (i.e., time post-transfection), cells were washed three times with phosphate-buffered saline (PBS) and then fixated using absolute methanol at room temperature for 5 min. After three washes with PBS, unspecific binding antigens were blocked with 1% bovine serum albumin (Sigma-Aldrich, St. Louis, MO, USA) for 1 h at room temperature. Staining with anti-FLAG primary antibody (1:100 Cell Signaling, Danvers, MA, USA) was carried out at room temperature for 1 h in the dark. Again, after three washes with PBS, the slides were incubated with the respective secondary antibodies (Thermo Fischer Scientific, Waltham, MA, USA) for 45 min in the dark at room temperature. Finally, cells were washed and then mounted with a fluorescence mounting medium (Dako, Real Carpinteria, CA, USA). Images were acquired using the 100× objective Nikon confocal Eclipse 80I microscope (Nikon Instruments Inc., Konan, Minato-ku, Tokyo, Japan). We followed the methodology described previously [23].

### 2.4. The Development and Optimization of ACE2 Maturation Assay

HEK293T cells obtained from ATCC were seeded in 12-well plates, and at approximately 90% confluency, they were transfected with wild-type ACE2 using ViaFect transfection reagent (Promega, Deira/Dubai, United Arab Emirates, Cat #E4981) in 500 μL media. Two hours post-transfection, cells were treated with either the tested drug or with their respective solvent mixed as control (mostly DMSO) with 500 μL media for a 1 mL final volume with the desired final concentration. After six hours of treatment (i.e., 8 h post-transfection), cells were harvested, and lysed with RIPA lysis buffer (Sigma) supplemented with a protease inhibitor cocktail (Sigma fast protease inhibitor cocktail), and the protein lysates were collected and quantified by the colorimetric bicinchoninic acid protein assay (Pierce™ BCA Protein Assay Kits, Thermo Fischer Scientific, Cat#23225). Glycosylation endoglycosidase H (Endo H) sensitivity and resistance assay were carried out using the EndoH digestion Sigma kit (Sigma, cat# A0810), and the glycosylation status was used to evaluate ACE2 maturation. This assay is routinely used to discriminate between fully N-glycosylated and immature glycoproteins [23,24,25,26,27,28]. Cell lysates were denatured in denaturation buffer (Sigma, cat# S4927) at 100 °C for 5 min, and all the subsequent steps were carried out according to the manufacturer’s protocol. Equal amounts of the proteins were incubated at 37 °C for 3 h in the presence or absence of 10 units of endoglycosidase H. About 15 µg of total protein lysates were resolved on 4–12% SDS-PAGE precast gradient gel (GeneScript, Rijswijk, The Netherlands), followed by a semi-dry transfer to a PVDF membrane. After blocking of unspecific binding antigens with 1% bovine serum albumin (Sigma-Aldrich) for 1 h at room temperature, membranes were incubated overnight with the primary antibodies: ACE2 Recombinant Rabbit Monoclonal Antibody (1:400 Invitrogen, Carlsbad, CA, USA, cat# PIMA532307) and anti-βactin (1:1000 Santa Cruz Biotechnology, Dallas, TX, USA, cat# sc-47778). Then, they were incubated with their corresponding secondary antibodies (Sigma-Aldrich) for 1 h at room temperature after the blots were washed three times with 1xTBST (Tris-buffered saline with 0.1% Tween^®^, Thermo Fischer Scientific). Membranes were then incubated with Enhanced Chemiluminescence Plus reagent (Pierce) and developed using the Typhoon FLA 9500 imager (GE Healthcare Bio Sciences, Piscataway, NJ, USA). Membranes were otherwise incubated with SuperSignal™ West Pico PLUS Chemiluminescent Substrate (Thermo Fischer Scientific, cat# 34579) and developed using the Azure Imaging Systems (Azure Biosystems Inc., Dublin, CA, USA). Western blots were analyzed, and the quantification was performed using ImageJ 1.53e software. The evaluation of the compounds was carried out in triplicates. The same process was conducted for the untreated (solvent-treated) samples.

### 2.5. ACE2 Maturation Rate Calculation

The protein bands (i.e., the mature and immature bands at ~120 kDa and ~75 kDa, respectively) detected by anti-ACE2 antibodies were quantified using ImageJ. Untreated samples were always used alongside the compounds tested samples in each experiment to minimize variability between experiments. The calculated maturation level of ACE2 was defined as the proportion of the total protein that was present in the mature band. It was calculated by inputting the bands’ intensities into the following equation:(Mature band/(mature + immature)) × 100

### 2.6. Statistics

We performed a parametric statistical assay and Welch’s *t*-test of two independent samples to determine if their populations’ means differed significantly at a significance level of 0.05. Statistics were analyzed to compare the means of two groups: the drug-treated group against the solvent-alone-treated group. Groups were made up of three replicas (N = 3). A *p* value < 0.05 is indicated by (*), a *p* value < 0.055 is indicated by (**), and a *p* value < 0.005 is indicated by (***).

### 2.7. The Development of ACE2 Maturation Assay

In order to develop an assay for the determination of the maturation level of ACE2 that can be used to evaluate the effects of different compounds and drugs, we transfected HEK293T cells with the FLAG-tagged human ACE2 wild-type (WT) in a mammalian expression plasmid (NM_021804; OriGene RC208442) and then performed a time-course evaluation of both the protein expression and ACE2 maturation levels. Before starting the project, we evaluated both the anti-ACE2 and the anti-FLAG antibodies for their specificity and sensitivity (i.e., ACE2 level detection), and we found that the anti-ACE2 antibodies (1:400 Invitrogen, cat# PIMA532307) were more specific and more sensitive and were able to detect lower levels of the expressed ACE2 protein. For example, the anti-FLAG antibodies were unable to detect the protein at the early expression time point (i.e., 4 h post-transfection), whereas the anti-ACE2 antibodies did. Note that we tested none transfected Cell lines for naturally expressed ACE2, in addition to Hek293T post ACE2 transfection at different time points and found ACE2 only in transfected cells after at least 4 h post transfection (Appendix A). Transfection agent used is FUGENE-HD. ACE2 construct used is OriGene RC208442. B-actin as loading control. GFP as transfection control. Therefore, we used the anti-ACE2 antibodies for all the subsequent experiments. The maturation level of ACE2 was determined by measuring the fraction of the protein that is present at the higher molecular weight protein band (~120 KDa) that is also resistant to digestion by Endo H. This Endo H resistance is indicative that the protein has passed from the ER to the plasma membrane through the Golgi complex and consequently received complex N-glycosylation moieties that cannot be cleaved off by Endo H. We observed that ACE2 seems to traffic and mature relatively quickly, with over 50% of the expressed protein in the mature state (~120 KDa protein band) within 4–5 h post-transfection as evidenced by being Endo H resistant and therefore mature (Figure 1A, note the 5 h time point). Based on the results of several optimization experiments and quantifications, we generated a graph showing the level of ACE2 maturation over time (Figure 1B). From this graph, we reasoned that 8–12 h post-transfection was a suitable time point for our screening assay because enough protein was being expressed for detection by western blotting with a significant proportion (~30%) of the detectable protein in the immature form and ~70 in the mature form (Figure 1A). It is worth noting that approximately 50% of the expressed protein was in the mature form even at a 5 h time point post-transfection, suggesting fast maturation of ACE2. We wanted an early time point where ACE2 synthesis and maturation were not saturated to avoid the accumulation of excessive amounts of the mature protein over the longer expression time points observed previously [23]. Figure 1A is an SDS-PAGE western blot of a representative experiment showing that the expressed ACE2 protein is detectable at the 5 h (post-transfection) time point onward and that over 50% of the protein is in the mature state as evidenced by its resistance to Endo H treatment. The quantifications of several experiments were performed and are presented in Figure 1B. These show that the maturation increased with time but was still in the linear range at the 8 h post-transfection. Therefore, we decided to use this time point for future experiments in our screening assay to evaluate the effects of the potential maturation inhibitors.

We wanted to further confirm ACE2 maturation by detecting its plasma membrane localization over time, as determined by confocal immunofluorescence microscopy, as a qualitative indicator of its maturation (Figure 2). Obviously, there was no expression of ACE2 or GFP-H-Ras (used as a plasma membrane marker) at the zero time point, but they both were clearly visible at the 4 h time point and beyond. The majority of the two proteins appear to co-localize at the plasma membrane within hours of co-expression. However, at the 4 h time point, some of the ACE2 seemed to be more intracellular, whereas the GFP-H-Ras was largely located at the plasma membrane.

## 3. Results

### 3.1. Brefeldin A Disrupted ACE2 Maturation While the Other Trafficking Modulator Showed Modest Effects

As a positive control for the total inhibition of ACE2 maturation, we used Brefeldin A (BFA), a well-established and drastic inhibitor of protein trafficking from the ER to the Golgi complex. This treatment showed total inhibition of the ACE2 protein maturation as indicated by the absence of the higher molecular weight 120 KDa band on SDS-PAGE gels and the quantitative presence of the lower molecular weight ~75 KDa band, which is presumably the immature protein (Figure 3A). This was confirmed by the quantitative cleavage of its N-glycans side chains, leading to a complete shift to a lower molecular weight protein band (Figure 3A). The level of maturation of ACE2 in cells treated with BFA was compared to those treated with DMSO. The latter showed >60% maturation within the 8 h of expression post-transfection, while the BFA treatment, as expected, showed no maturation at all (Figure 3B).

Small molecular modulators of protein trafficking and transport usually exhibit general effects. Figure 4A shows the effects of some selected modulators in a bar graph in comparison with BFA, which shows a complete blockage of ACE2 maturation. Exo1 inhibits ER to Golgi trafficking without affecting the exosomal compartments [29]. We tested the effect of this compound at 20 μM on ACE2 maturation, and it showed a moderate but significant decrease in the amount of mature ACE2 (Figure 4A,C). Similarly, AG1478, which selectively targets the cis-Golgi without affecting the endosomal compartment [30], caused a significant reduction in ACE2 maturation (Figure 4D). Similar results were also obtained with Casin at 8 and 5 μM (Figure 4E and Figure 4F, respectively); Casin is a Cdc42 GTPase inhibitor affecting Golgi to plasma membrane trafficking [31]. On the other hand, Berbamine, an inhibitor of the TRPMLs-mediated transport causing the inhibition of ACE2 endolysosomal trafficking [32], did not show any effects at the tested concentration (Figure 4G).

### 3.2. Evaluating the Effects of Some Reported Drugs Affecting SARS-CoV-2 Virus or COVID-19 Severity Demonstrated Modest Impacts on ACE2 Maturation

Several compounds have been reported in the literature to affect COVID-19 severity. Since the effect of S protein on acute lung injury depends on ACE2 [33] and high ACE2 expression associated with severe COVID-19 symptoms [34]. Therefore, we wanted to evaluate if their efficacy in impacting the disease might be attributed to their potential impact on ACE2 maturation. It has been reported that histone deacetylase inhibitors such as Panbinostate have the potential to function as a preventative agent against COVID-19 [35]. We therefore evaluated the effects of Panbinostate at 10 μM, which caused a significant decrease in ACE2 maturation (Figure 5B). In addition, Sodium Butyrate at 1 mM also caused a significant reduction (Figure 5C). Both Panbinostate and Sodium Butyrate are HDAC inhibitors that suppress the expression of the *ABO* gene, which encodes A-transferases and B-transferases [35]. On the other hand, SAHA, another HDAC inhibitor, which was reported to suppress ACE2 expression [35], showed no effects at ACE2 maturation at 3.5 μM using our assay (Figure 5D).

We also tested Celastrol at 0.75 μM, a lead compound that inhibits SARS-CoV-2 replication possibly by inhibiting the activity of viral and human cysteine proteases [36], and CK869 at 100 μM, an inhibitor of the actin-related protein 2/3 (ARP2/3) complex that was reported to downregulate ACE2 expression in cells [37,38]. They did not show any effects on ACE2 maturation using our assay (Figure 5E and Figure 5F, respectively).

Furthermore, we evaluated several other compounds that resulted in significant reductions in ACE2 maturation, including Afatinib, a kinase inhibitor of HER2 and EGFR [38,39], at 30 μM (Figure 5G), and EIPA ([5-(N-ethyl-N-isopropyl)-amiloride], an inhibitor of the Na^+^/H^+^ exchanger, at 40 μM (Figure 5H).

Small decreases in ACE2 maturation have been observed with paracetamol at 11 mM, Chloroquine Diphosphate at 90 μM (Figure 5I and Figure 5J, respectively). Surprisingly as low as 200 nM of Cholecalciferol (Vitamin D3) was able to cause a significant decrease in ACE2 maturation (Figure 5K). In contrast, both methyl-β-cyclodextrin at 2 mM and LY294002 at 15 μM showed enhanced ACE2 maturation (Figure 5L and Figure 5M, respectively).

### 3.3. ACE2 Maturation Is Not Influenced by Statins

Statin therapy, a widely used class of medications used for the treatment of hypercholesterolemia, was reported to positively influence disease outcomes in COVID-19 patients, with early observations suggesting beneficial impacts on clinical outcomes [40,41,42]. That reported impact could be attributed to their anti-inflammatory, anti-thrombotic, and immuno-modulatory effects. They may also influence viral entry into human cells [43]. It was reported by Zapatero-Belinchón et al. that “Fluvastatin was the only statin found to inhibit low and high pathogenic coronaviruses in vitro and ex vivo in a dose-dependent manner” [44]. Therefore, we wanted to examine if statins could influence ACE2 maturation but found no detectable impact at the tested concentrations for all the tested statins, including Fluvastatin, Mevinolin (Lovastatin), Rosuvastatin, Simvastatin, Pravastatin and Atorvastatin (Figure 6A,B). The western blot data for Fluvastatin are shown in Figure 6C as an illustrative example.

## 4. Discussion

Small molecular modulators of protein trafficking and cellular transport were selected for testing as potential inhibitors or modulators of ACE2 maturation. Some of the selected compounds inhibited certain steps within the secretory pathway along the route from ER to the plasma membrane (i.e., its site of synthesis to its site of action), as shown in Figure 7. As expected, BFA showed a total block of ACE2 maturation, but other trafficking modulators, such as Casin, a Cdc42 GTPase inhibitor affecting Golgi to PM trafficking [31], showed a concentration-related slight reduction in ACE2 maturation at 8 μM and 5 μM (Figure 4E,F). On the other hand, Berbamine at 10 μM showed a limited or no effect (Figure 4G). Berbamine may compromise the endolysosomal trafficking of ACE2 by inhibiting TRPMLs [32]. In fact, Berbamine has been suggested to prevent SARS-CoV-2 from entering the host cells by increasing the secretion of ACE2 by exosomes, resulting in decreased levels of ACE2 at the cell surface [32,45].

Based on published literature, we screened a selected group of drugs and compounds to uncover potential leads for ACE2 maturation inhibitors that may work against coronavirus entry. Screened drugs that caused significant reduction in ACE2 maturation are represented in a diagram in the Appendix A. Consequently, these may act as potential treatments for COVID-19 and related coronavirus infections. It is suggested in the literature that decreased availability of the viral receptor might give some protection against cycles of cellular infections [46]. However, as shown in Figure 8, ACE2 plays a role in inflammation progression [47]. In this scenario, we should take into consideration that elevated levels of angiotensin II could be generated and bind to AT1 receptors, resulting in vasoconstriction, enhanced inflammation, and thrombosis [48]. However, membrane-bound and soluble ACE2 limit those detrimental effects through the degradation of angiotensin II, which results in increased generation of angiotensin1-7 that triggers counter-regulatory protective effects through binding to G-protein coupled Mas receptors [47].

Through our screening assay, we showed that some COVID-19 drugs were able to mildly reduce the maturation level of ACE2 (Figure 5A). Decreased maturation rates are expected to increase the time required by ACE2 to reach its final destination, which is the plasma membrane. This is expected to result in reduced functional ACE2 on the cell surface and, consequently, reduced angiotensin II cleavage. Actually, it has been reported that acute pulmonary inflammation and coagulation are medical complications that arise in response to enhanced and unopposed angiotensin II detrimental effects [49]. As reported in several clinical case studies of patients infected with SARS-CoV-2, the infection level and the severity of the disease are both associated with several risk factors, including older age, diabetes, hypertension, and cardiovascular diseases, which could be attributed to reduced ACE2 levels since all share a variable degree of ACE2 deficiency [49,50,51]. If an additional ACE2 reduction occurs after the COVID-19 treatment, this might amplify the dysregulation of the detrimental effects of the angiotensin II and AT1 receptor axis [49]; see the chart presented in Figure 8. For example, although a retrospective analysis showed that paracetamol was associated with decreased ACE2 protein expression and a lower risk of COVID-19 infection [52], on the other hand, a clinical study analyzing paracetamol as an early COVID-19 treatment indicated it had caused an increase in the risk of hospitalization [53]. This may indicate that the outcome might be infection-stage dependent. Therefore, we suggest that ACE2 down-regulation induced by the treatment of choice might be detrimental in people with baseline ACE2 deficiency associated with the above conditions.

Paracetamol in our screening showed a significant decrease in ACE2 maturation (Figure 5I). which may explain the finding of a retrospective analysis that identified a putative protective effect of paracetamol against SARS-CoV-2 infection, suggesting that “paracetamol intake was associated with a lower occurrence of COVID-19”. However, it was also reported that paracetamol resulted in decreased ACE2 protein levels in cultured cells [52], which is in agreement with our results showing a small but significant decrease in ACE2 maturation (Figure 5I). This may indicate that we should take into account both the degree of infection and how much ACE2 is available to neutralize the stress induced on the body due to inflammation, all to prevent the COVID-19 patient from getting into worse medical complications due to elevated angiotensin II levels.

Statins, commonly used drugs to treat hyperlipidemia by blocking cholesterol synthesis [54], showed no effects on ACE2 maturation in our screening (Figure 6A,B), while other chemical inhibitors, such as methyl-β-cyclodextrin, which removes cholesterol from plasma [55] showed slightly improved ACE2 maturation (Figure 5L); meaning that ACE2 levels are maintained normal and are not subjected to reduced availability. In a meta-analysis of retrospective observational studies, it was shown that “statin therapy was associated with an about 35% decrease in the adjusted risk of mortality in hospitalized COVID-19 patients” [42]. In a nationwide Swedish cohort study, “Statin-treated individuals appear to have lower COVID-19 mortality than nonusers, whether assessed in the general population, from COVID-19 onset or from hospitalization” [40]. Although pieces of evidence are present for ACE2’s protective role, statins reported effects might be through other pathways.

Chloroquine is an antimalarial drug that has been used as a COVID-19 treatment [56]. It was found to increase the endosomal pH and affect the glycosylation of ACE2 [57] because glycosylation is a vital step for ACE2 maturation, which may explain the small but significant decrease in ACE2 maturation found in our assay for this drug (Figure 5J).

ACE inhibitors were not included in our screening because they reportedly increase the levels of ACE2 [21,58,59,60]. In regards to COVID-19, ACE inhibitors could reduce SARS-CoV-2 cell entry by reducing the availability of binding sites and reducing internalization of ACE2. However, it should be taken into consideration that a virus only needs one receptor to infect a cell.

## 5. Conclusions

The cellular screening assay was developed and optimized to evaluate the trafficking and maturation levels of ACE2 in response to treatments with several drugs and compounds. Some have been suggested for consideration as potential COVID-19 treatments. Our screening revealed the relatively fast and robust ACE2 maturation, which was only mildly affected by certain drug and compound treatments with varying degrees. However, out of the twenty-three tested compounds, eight showed a limited but significant decrease in ACE2 maturation levels with a *p* value less than 0.05, three showed a ~20% decrease, while Brefeldin A, as expected, showed a complete block of ACE2 maturation. To conclude, screening of ACE2 trafficking inhibitors showed a significant effect on most molecular modulators of transportation and mild effects on most COVID drugs. No effects were observed for statins. Targeting ACE2 for the manipulation of its bioavailability might be beneficial in certain scenarios and harmful in others. Therefore, using this as a COVID-19 therapeutic strategy requires detailed research to better understand its biology and impact on some viral infections.

## Figures and Tables

**Figure 1 biomolecules-14-00764-f001:**
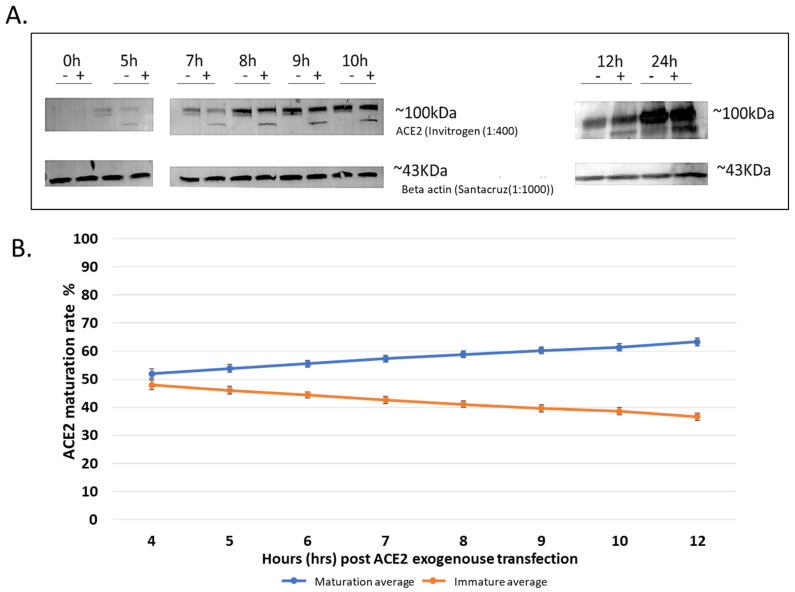
Maturation of exogenously expressed ACE2 over time. Hek293T cells transfected with ACE2 and collected after different time points (0, 5, 7, 8, 9, 10, and 12, then finally 24 h) (**A**); Representative western blots which resulted from 30 µg of digested cell lysates, loaded on a BioRad precast gel. The enzymatic Endo H digestion cleaves off the carbohydrate side chains from the immature glycoprotein, creating a band shift to a lower molecular weight size band. ACE2 mature and immature bands are detected with (Invitrogen) specific antibodies, showing the two bands at ~120 kDa and ~75 kDa. To control for variations, including loading variation, we always included a control group of samples processed in parallel with the samples for the tested drug. In addition, to control for the loading, the level of ACE2 maturation is always in relation to the total expressed ACE2. Beta-actin was used as a loading control. Original images can be found in Appendix A. (**B**) showing a trend line over time. Experiments were conducted three times independently, and the graph shows the average of three replicas; quantification of bands for ACE2 at different hrs-post-trans (hours post-transfection) was done using ImageJ.

**Figure 2 biomolecules-14-00764-f002:**
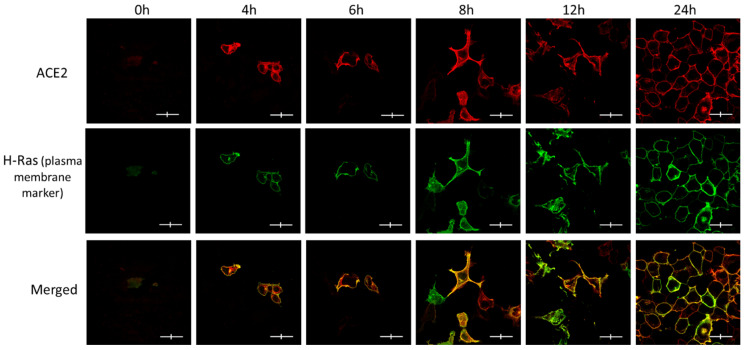
The subcellular localization of ACE2 in Hela cells at the various post-transfection time points and its co-localization with the plasma membrane marker (GFP-HRas). Immunofluorescence images show a time-dependent increase in the signals for both ACE2 (red) and GFP-HRas (green). Cells were co-transfected with ACE2 WT and the plasma membrane marker as described in the methods. Slides were fixed and probed and then visualized using the fluorescence confocal microscope. Images were captured with (Nikon Eclipse i80, Tokyo, Japan) at ×100 magnification power. Scale bar 20 μm × 4 μm.

**Figure 3 biomolecules-14-00764-f003:**
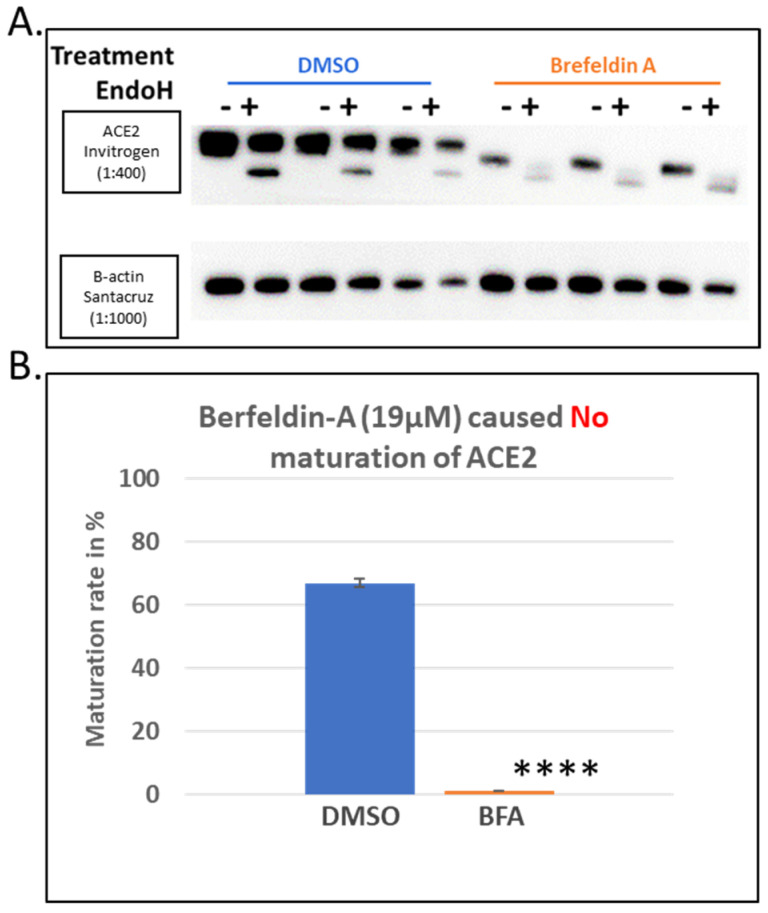
Brefeldin A (BFA) as a positive control for the screening assay of ACE2 maturation inhibitors. Hek293Tcells transfected with ACE2 WT with ViaFect transfection reagent, then 2 h post-transfection, cells were treated with a final concentration of 19 μM of Brefeldin A dissolved in DMSO. As indicated earlier in the methods, control groups of samples (treated with vehicle solvent only) were processed in parallel with their respective treated group, and both groups were processed 8 h post-transfection. (**A**). Western blot comparing the BFA-treated samples with those treated with the solvent alone (DMSO) and harvested 6 h post-treatment. N = 3. Original images can be found in Appendix A. (**B**). A bar graph showing the quantified average maturation levels of each treatment (BFA versus DMSO), as described in the methods section. Bands were quantified using ImageJ software. Statistics: a two-tailed student test (Welch’s test) was applied (“****” indicate *p* value < 0.005).

**Figure 4 biomolecules-14-00764-f004:**
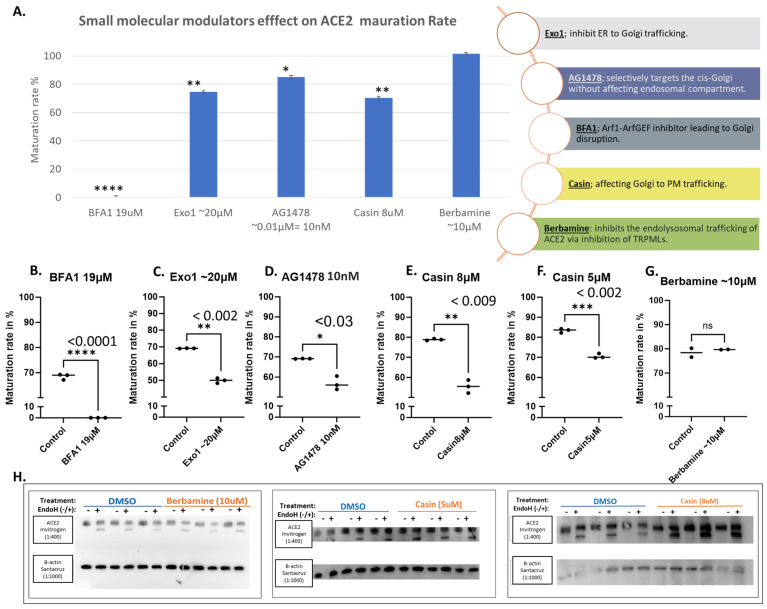
Selected small molecular modulators of transportation effects on ACE2 maturation. Hek293Tcells transfected with WT-ACE2 by ViaFect transfection reagent, then 2 h post-transfection cells were treated with varying final concentrations as indicated on the *x*-axis and then culture for another 6 h (i.e., 8 h post-transfection). Control groups were processed alongside their respective treatment groups and processed in a similar fashion. Maturation rates calculated are for both bands in the same lane, yielding measurements of the maturation percentages regardless of equal loading between lanes. (**A**). A bar graph shows the average maturation of each group, following the equation in the methods section. All maturation levels are normalized against the control group. (**B**–**G**). Maturation levels for both treated and control groups are displayed as individual readings and plotted with the mean difference tested for significance using GraphPad 5. (**H**). Western blots showing the Berbamine (10 μM) treated group and Casin (5 μM or 8 μM) both compared against the solvent-alone control (DMSO) group, all were harvested 6 h post-treatment. N = 3. Original images can be found in Appendix A. Bands were quantified with ImageJ software. Statistics: unpaired two-tailed student *t*-test (Welch’s *t*-test) was applied; **** (*p* value < 0.005), *** (*p* value < 0.002), ** (*p* value < 0.055), * (*p* value ≤ 0.05) and ns: not-significant.

**Figure 5 biomolecules-14-00764-f005:**
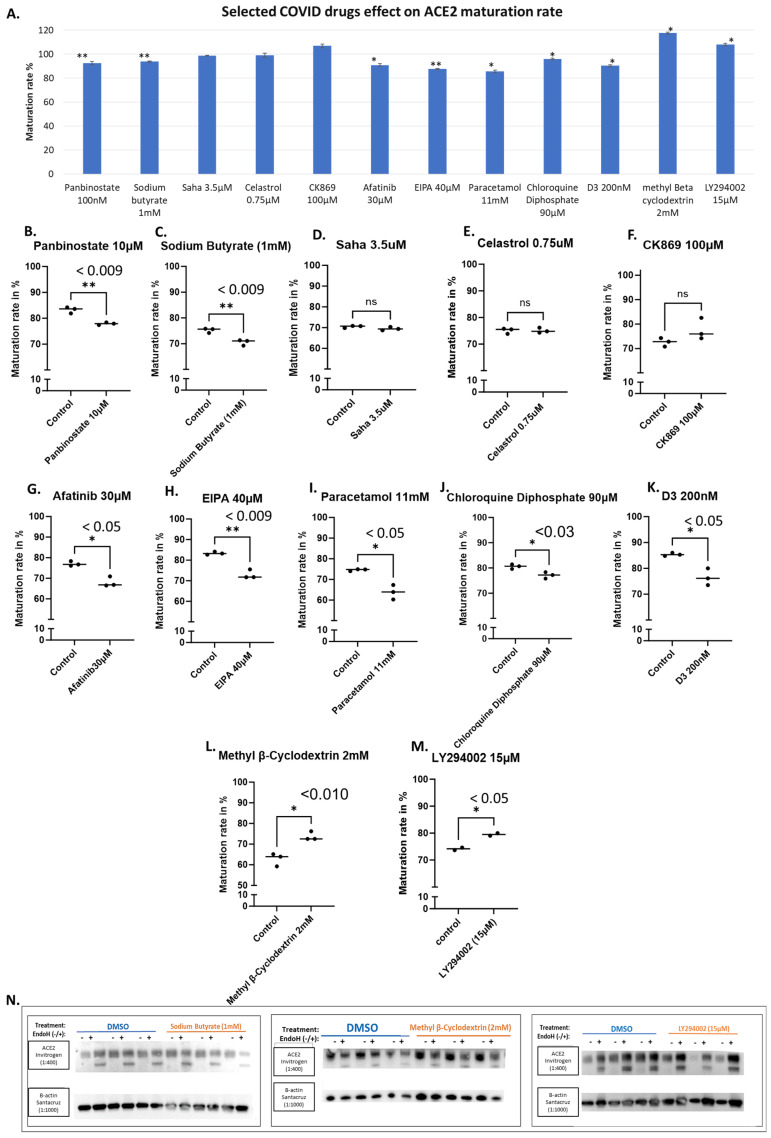
Selected COVID-19 drugs and their effects on ACE2 maturation. HEK293T cells transfected with WT-ACE2 with ViaFect transfection reagent, then 2 h post-transfection cells were treated with varying final concentrations as indicated on the *x*-axis and then incubated for another 6 h. Control groups were processed alongside their respective treated groups and were processed by the same steps and at the same time. Maturation levels were determined by calculating the quantities of the two bands in each lane. (**A**). A bar graph shows the average maturation level of each group, following the equation in the methods section. All maturation levels were normalized against the control group. (**B**–**M**). Maturation levels for both treated and control groups are displayed as individual readings and plotted with the mean difference tested for significance using GraphPad. (**N**). Western blots showing either Sodium Butyrate (1 mM), methyl Beta cyclodextrin (2 mM), or LY294002 (15 μM) and compared against the solvent-alone control (DMSO) group, all were harvested 6 h post-treatment. N = 3. Original images can be found in Appendix A. Bands were quantified using ImageJ software. Statistics: unpaired two-tailed student *t*-test (Welch’s *t*-test) was applied; ** (*p* value < 0.055), * (*p* value ≤ 0.05) and ns: not-significant.

**Figure 6 biomolecules-14-00764-f006:**
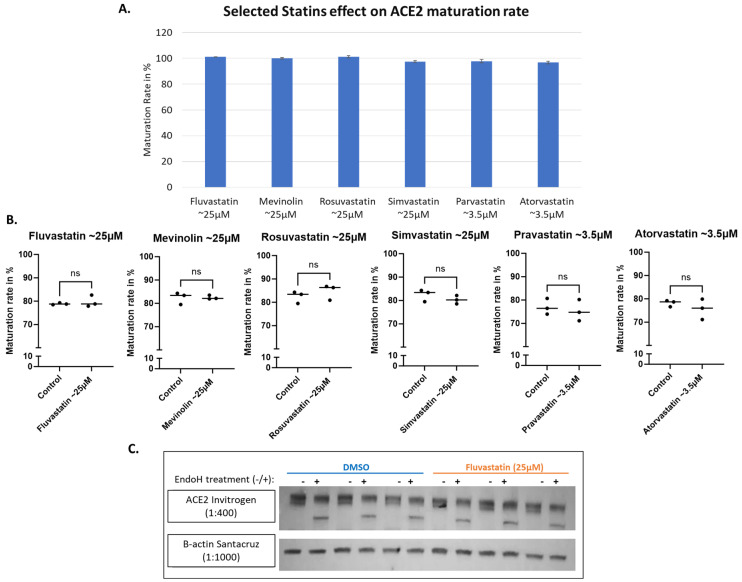
Selected group of statins all show no significant effect on ACE2 maturation. HEK293T cells transfected with WT-ACE2 with ViaFect transfection reagent (Promega, cat #E4981), then 2 h post-transfection cells were treated with varying final concentrations of statins as indicated on the *x*-axis then incubated for another 6 h. (**A**). A bar graph shows the average maturation level of each group, following the equation in the methods section. All maturation levels were normalized against the control group. (**B**). Maturation levels for both treated and control groups are displayed as individual readings and plotted with the mean difference tested for significance using GraphPad. (**C**). Western blot showing Fluvastatin (25 μM) compared against the solvent alone as a control (DMSO) group. All were harvested 6 h post-statin treatment. N = 3. Original images can be found in Appendix A. Bands were quantified using ImageJ software. Statistics: unpaired two-tailed student *t*-test (Welch’s *t*-test) was applied; ns: not-significant.

**Figure 7 biomolecules-14-00764-f007:**
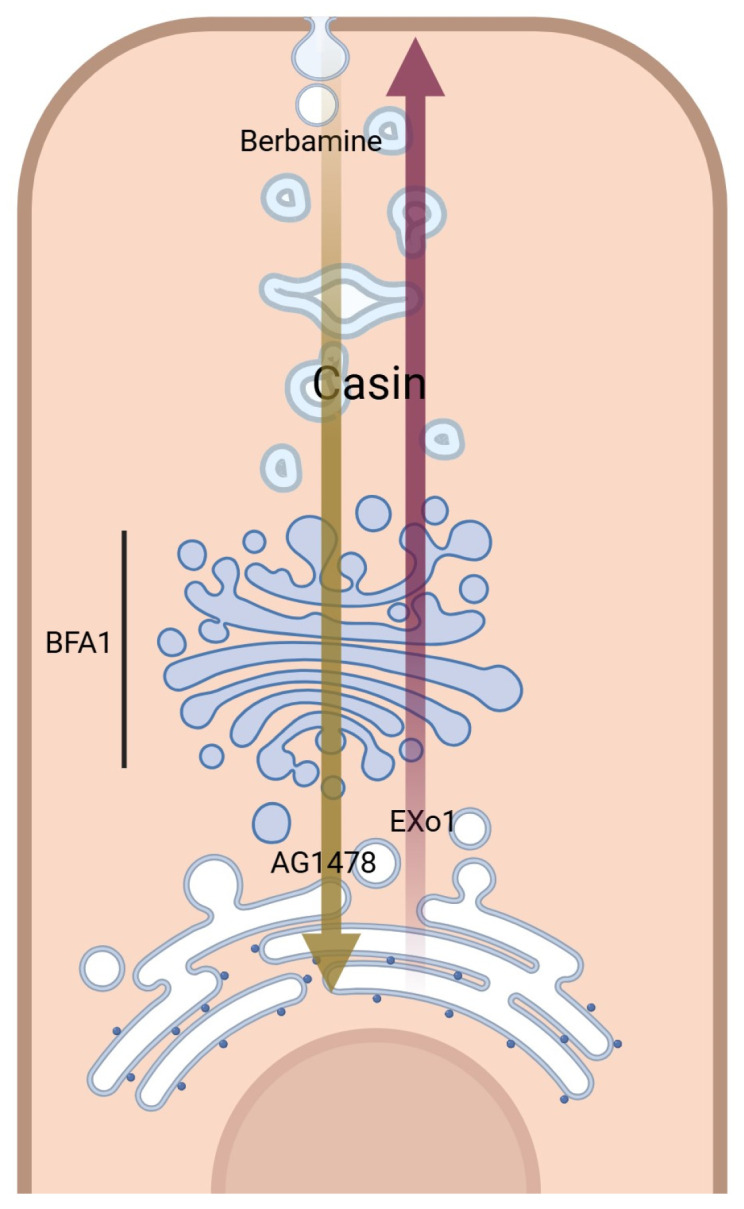
A schematic representation of small molecular modulators of transportation and their corresponding sites of action (Created with BioRender.com, accessed on 10 June 2024, license are acquired).

**Figure 8 biomolecules-14-00764-f008:**
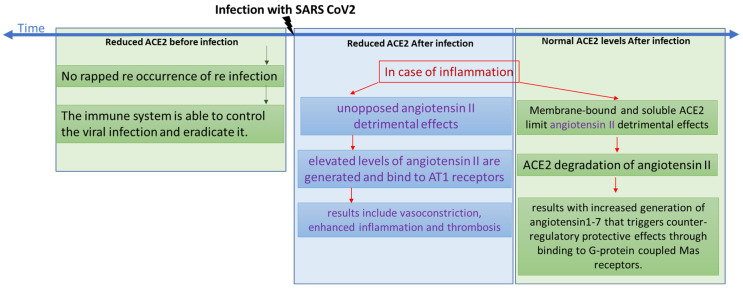
A chart showing the different scenarios with regards to time of infection compared with levels of ACE2.

**Table 1 biomolecules-14-00764-t001:** Drugs and compounds evaluated in this study with their sources and catalog numbers.

#	Class	Drug	Note	Catalog #	Supplier	Solvent Used
1	Small molecular modulators of protein trafficking	Brefeldin A	Potent inhibitor of protein trafficking from the ER to the Golgi apparatus	B6542	Sigma (St. Louis, MO, USA)	DMSO
2		Exo-1	Vesicular trafficking inhibitor	ab120292	Abcam (Cambridge, UK)	DMSO
3		AG 1478	EGF receptor tyrosine kinase inhibitor (the same as BFA but less cytotoxic)	ab141438	Abcam	DMSO
4		Casin	Rho GTPase Cdc42 inhibitor	SML12553	Sigma	DMSO
5		Berbamine	Natural STAT3 inhibitor	547190	Sigma	DMSO
6	COVID-19 investigated drugs	Panobinostat (LBH589)	HDAC inhibitor, suppresses ACE2 expression	sml3060	Sigma	DMSO
7		Sodium butyrate	HDACIs, suppresses *ABO* expression in the same way as panobinostat	303410	Sigma-Aldrich	DMSO
8		SAHA (Vorinostat in dimethyl sulfoxide form)	Histone deacetylase (HDAC) inhibitor	10009929	Cayman Chemical Company (Ann Arbor, MI, USA)	DMSO
9		Celastrol	Disrupts Hsp90/Cdc37 complete	34157-83-0	Sigma	DMSO
10		Ck869	An inhibitor of the actin-related protein 2/3 (ARP2/3) complex	C9124	Sigma	DMSO
11		Afatinib	A kinase inhibitor of HER2 and EGFR	S1011	Selleckchem (Cologne Germany)	DMSO
12		Eipa [5-(n-ethyl-n-isopropyl)-amiloride]	An inhibitor of the Na^+^/H^+^ exchanger (NHE)	14406	Cayman Chemical Company	DMSO
13		Paracetamol	Highly effective analgesic and antipyretic properties	P0300000	edQm (Strasbourg France)	DMSO
14		Chloroquine diphosphate	A DNA intercalator that inhibits cell growth and induces cell death	C6628	Sigma	H_2_O
15		Cholecalciferol (vitamin D3)	A total of 85% infectious virus reduction	740292	Sigma	DMSO
16		LY294002	an inhibitor of PI3K	L9908	Sigma	DMSO
17	Cholesterol-lowering drugs	Methyl-β cyclodextrin	Depolymerizes the actin cytoskeleton, HMG-CoA reductase inhibitor	C4555	Sigma	DMSO
18		Fluvastatin	Statins	SML0038	Sigma	DMSO
19		Lovastatin (mevinolin)	Statins Hypocholesterolaemia drug HMG-CoA reductase inhibitor	M2147	Sigma	DMSO
20		Rosuvastatin	Statins	SML1264	Sigma	DMSO
21		Simvastatin	Statins	S6196	Sigma	DMSO
22		Pravastatin	Statins	P4498	Sigma	H_2_O
23		Atorvastatin	Statins	PZ0001	Sigma	DMSO

## Data Availability

The original contributions presented in the study are included in the article/Appendix A, further inquiries can be directed to the corresponding author.

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
