# Peer review of "The Evaluation of Drugs as Potential Modulators of the Trafficking and Maturation of ACE2, the SARS-CoV-2 Receptor"

_biomolecules, 2024, doi:10.3390/biom14070764_

Round 1
Reviewer 1 Report
Comments and Suggestions for Authors
Please see the attached file.

The introduction is well-written. The discussion, though, needs to be edited.
Author Response
Please see the attachment for the updated manuscript with supplementary figures added. Kindly note: highlighted in yellow are added information to the methodology. and the discussion section was re-written.
Comment 1 The authors indicate that a mature ACE2 should be seen at a molecular weight of 125 KDa, versus an immature protein without complex N-glycosylation moieties that can be cleaved and would be seen at a molecular weight of 75KDa. While this assay seems valid, in my opinion, having another orthogonal method to strengthen the results would significantly elevate the impact of the manuscript.
Response: We thank the reviewer for all his constructive comments including this important one. Western blotting analysis using protein specific antibodies following SDS-PAGE separation is the most commonly used and largely accepted approach to determine the approximate sizes of proteins. In our experiments, we are using anti-ACE2 specific antibodies for detecting exogenously expressed ACE2 and as negative controls we always use lysates from un-transfected cells processed in parallel and we are therefore confident that the detected proteins bands are indeed ACE2 species. In addition, our results are in full agreement with the extensive literature on exogenously expressed ACE2. Furthermore, we postulated that the two protein bands appearing on our Western blots are the mature (fully N-glycosylated) and the immature (not-fully N-glycosylated) species of ACE2. To further confirm that, we carried out Endo H sensitivity and resistance assays (that includes Western blotting following Endo H digestion) with the higher molecular weight band (at ~120KDa) not changing mobility upon this treatment confirming its N-glycan maturation status while the mobility of the lower molecular weight protein band shifting to a lower position in the gel confirming our assumptions that it is the immature form of Ace2. We hope that this reviewer will accept our confidence on the validity of the assay to identify of the expressed ACE2 species and that Endo H analysis as confirmatory assay.
Comment 2 "The discussion, though, needs to be edited. It is hard to follow all the biological mechanisms related to each drug and to understand what the author’s conclusions are. A schematic figure could be beneficial."
Response: Thank you again for this suggestion. Although it is hard to provide one diagram to cover all the tested drugs, we are now including the following diagrams (in supplementary as figure2) that focus on the postulated mechanisms of the compounds that showed decreased ACE2 maturation only. Highlighting the importance of ACE2 levels pre-infection Vs. post-infection. Another two were provided in the discussion section as figure 7 and figure 8 providing clarity to the site of action for the selected small molecular modulators of transportation and the invaluable role of ACE2 level Vs. time of infection for COVID19 patients.
Comment 3 Section 2.4- The authors use HEK293T cell line and should include a blot showing that this cell line does not express ACE2, or at least mention this with supporting references. The same comment for HELA cell line.
Response: In all our experiments we always include un-transfected cells or transfected cells harvested at zero time point and in all those cases, lysates from those cells are included in our Western blotting analysis as negative controls and showing no protein bands at the expected size including when probed with ACE2-spscific antibodies. Please note the zero-time point in figure 1, as well as, the transfected Cell lines and early-time-points post-transfection; in the new supplementary-figure (1).
Comment 4 Section 2.7- The authors transfected the cells with FLAG-tagged ACE2. Why don’t they use an anti-FLAG antibody for these blots, but rather use anti ACE2 antibody?
Response: At the start of our project, we evaluated both the anti-ACE2 and the Anti-FLAG antibodies for their specificity and sensitivity (i.e. ACE2 level detection) and we found that the anti-ACE2 antibodies were more specific and more sensitive and were able to detect lower levels of the expressed ACE2 protein. For example, the anti-FLAG antibodies were unable to detect the protein at the early expression time point (i.e.4 hours) whereas the anti-ACE2 antibodies did and we therefore decided to use anti-ACE2 antibodies for subsequent experiments especially that HEK293T and other cell lines did not show any endogenous ACE2 expression (Supplementary-figure 1). We are now indicating this in the methods (lines 173-178).
Comment 5 Do you expect the bands to include the multiple isoforms of ACE2?
Response: The expressed protein is derived from the cDNA of isoform 2 of ACE2 (Uniprot ID Q9BYF1) cloned into the used mammalian expression vector (NM_021804; OriGene RC208442) and we therefore do not expect to have expression of the other ACE2 isoforms. We are including this information in the relevant methods section and added the Uniprot ID (lines 113-115).
Comment 6 The authors indicate that the mature state is evidenced by ACE2 resistance to Endo H treatment. I understand that theoretically, Endo H would not cleave complex glycans, but please add a control to prove that, as the whole analysis relies on this assumption.
Response: The Endoglycosidase H sensitivity and resistance assay is well established and accepted as discriminatory assay between mature and immature N-glycans of glycoproteins that pass through the secretory pathway. We are now including references to the relevant literature (lines 134-135) including some of our own published work demonstrating the use of this assay on other proteins which can be considered as controls (Ali et al. 2011; Badawi et al. 2022; Freeze and Kranz n.d.; Gariballa, Badawi, and Ali 2024; Hume et al. 2009; Wu and Ertelt n.d.). In addition, the shift in the mobility of the lower molecular weight band (but not the higher molecular weight band) confirms our assumptions.
References:
Ali, Bassam R. et al. 2011. “Endoplasmic Reticulum Quality Control Is Involved in the Mechanism of Endoglin-Mediated Hereditary Haemorrhagic Telangiectasia.” PLOS ONE 6(10): e26206. https://journals.plos.org/plosone/article?id=10.1371/journal.pone.0026206 (June 8, 2024).
Badawi, Sally et al. 2022. “Characterization of ACE2 Naturally Occurring Missense Variants: Impact on Subcellular Localization and Trafficking.” Human Genomics 16(1). https://doi.org/10.1186/s40246-022-00411-1 (March 5, 2024).
Freeze, Hudson H, and Christian Kranz. “Endoglycosidase and Glycoamidase Release of N-Linked Glycans.” http://www.functionalglycomics.org (June 8, 2024).
Gariballa, Nesrin, Sally Badawi, and Bassam R. Ali. 2024. “Endoglin Mutants Retained in the Endoplasmic Reticulum Exacerbate Loss of Function in Hereditary Hemorrhagic Telangiectasia Type 1 (HHT1) by Exerting Dominant Negative Effects on the Wild Type Allele.” Traffic 25(1): e12928. https://onlinelibrary.wiley.com/doi/full/10.1111/tra.12928 (June 8, 2024).
Hume, Alistair N. et al. 2009. “Defective Cellular Trafficking of Missense NPR-B Mutants Is the Major Mechanism Underlying Acromesomelic Dysplasia-Type Maroteaux.” Human Molecular Genetics 18(2): 267–77. https://dx.doi.org/10.1093/hmg/ddn354 (June 8, 2024).
Wu, Zhengliang L, and James M Ertelt. “Endoglycosidase Assay Using Enzymatically Synthesized Fluorophore-Labeled Glycans as Substrates to Uncover Enzyme Substrate Specificities.” https://doi.org/10.1038/s42003-022-03444-3 (June 8, 2024).
Comment 7 What are the percentages of DMSO used? Please also indicate the DMSO % in Table 1, as DMSO is known to inhibit the interaction between ACE2 and S1 at concentrations above 5% and to inhibit other enzymes.
Response: The DMSO treatment was used to rule out solvent effects. The volumes of DMSO used as a control for each experiment was always equivalent to the volume used for the drug treated sample. In all cases, the percentage of DMSO never exceeded 0.4% and we are now indicating that in the methods (lines 104-107).
Comment 8 " How were the concentrations of the drugs selected? Please add references to support the selected concentrations. How do the authors know that the concentrations used are enough to identify an effect? The authors should do a titration experiment with several concentrations to conclude there is no effect, rather than trying a single dose."
Response: Prior to deciding on the concentration of each of the tested drugs, we carried out literature search on the tolerable concentration and in most cases tested multiple concentrations around what was found in the literature. We are now citing the relevant literature (new supplementary table 1) that we have used as the starting point for our optimization of the screening. We indicate that in the manuscript (lines 106-107).

Reviewer 2 Report
Comments and Suggestions for Authors
The research article titled “The Evaluation of Drugs as Potential Modulators of the Trafficking and Maturation of ACE2, the SARS-CoV-2 Receptor” is well conducted and presented.
The authors have developed a cellular assay to evaluate the trafficking and maturation levels of ACE2 and assessed the potential impact of 23 drugs and compounds on its maturation, presenting the data comprehensively. The introduction is well-written with relevant references, providing a solid foundation for the study.
The assay development and result presentation are executed effectively. However, the selection of drugs investigated for their impact on COVID-19 must include those that are either approved or currently in clinical trials. The drugs listed in Table 1 are not suitable for COVID-19 treatment at this stage. For instance, paracetamol is not a drug that cures COVID-19 infection, yet it was evaluated in this study.
For these reasons, I cannot recommend this article for publication in Biomolecules. I strongly suggest that the authors test their assay with appropriate antiviral drugs that are relevant to COVID-19 treatment.
Author Response
Please see the attachment for the updated manuscript with supplementary figures added. Kindly note: highlighted in yellow are added information to the methodology. and the discussion section was re-written.
Comments: The assay development and result presentation are executed effectively. However, the selection of drugs investigated for their impact on COVID-19 must include those that are either approved or currently in clinical trials. The drugs listed in Table 1 are not suitable for COVID-19 treatment at this stage. For instance, paracetamol is not a drug that cures COVID-19 infection, yet it was evaluated in this study. For these reasons, I cannot recommend this article for publication in Biomolecules. I strongly suggest that the authors test their assay with appropriate antiviral drugs that are relevant to COVID-19 treatment.
Response: The main stated objectives of the manuscript is to develop an assay for ACE2 trafficking and maturation and using it to screen and investigate the impact of a group of small modulators and drugs on the cellular trafficking and maturation of ACE2 as potential leads for identifying new drugs for COVID19 and not necessarily drugs to be used or with a proven impact of COVID19. This is clear from our statements such as "The cellular screening assay was developed and optimized to evaluate the trafficking and maturation levels of ACE2 in response to several drugs and compounds some are being considered as potential COVID19 treatments." and "Targeting ACE2 for the manipulation of its bioavailability might be beneficial in certain scenarios and harmful in others and therefore using this as a COVID-19 therapeutic strategy requires detailed understanding of its biology." Having said, this suggestion is an excellent one and we will use for our future work as we will utilize the described assay for further studies to screen more recently described potential COVID19 drugs. Unfortunately, results of such studies would require extended period of time that is beyond the scope and timeframe of this manuscript and review process. We hope that the reviewers and the editor are convinced by the validity of the assay on monitoring ACE2 maturation and its potential utility and contributions to the literature in the field of biomolecules, especially we are presenting the screening of over twenty molecules.

Round 2
Reviewer 1 Report
Comments and Suggestions for Authors
The authors pretty much answered my concerns.
Comments on the Quality of English Language
There are multiple spelling errors throughout the paper. For example, in Figure 1: "mturation avarage" , "SanraCruze" , "Invertogen". All of these errors are in one figure.
I strongly suggest that the paper be edited by a professional English editor.
1. Figure 3-
a. Legend- Please correct typo errors in line 5: “were proceeesd in parallel…”
This comment was written in the previous round of review and was not corrected.
Another example from Figure 4 legend:
"yealding accurate percentages ragrdless of equal loading between lanes".
Author Response
We thank the reviewer for highlighting this deficiency within the manuscript, the manuscript has been heavily edited as requested.

Reviewer 2 Report
Comments and Suggestions for Authors
Accept in present form
Author Response
Thank you for your time and consideration.